# Comparison of the Corrosion Behavior of Brass in TiO_2_ and Al_2_O_3_ Nanofluids

**DOI:** 10.3390/nano10061046

**Published:** 2020-05-29

**Authors:** Siyu Xie, Yi Zhang, Yanfang Song, Fang Ge, Xin Huang, Honghua Ge, Yuzeng Zhao

**Affiliations:** Shanghai Engineering Research Center of Energy-Saving in Heat Exchange Systems, Shanghai Key Laboratory of Materials Protection and Advanced Materials in Electric Power, Shanghai University of Electric Power, Shanghai 200090, China; siyuxie97@163.com (S.X.); zzzyyy1997@163.com (Y.Z.); yanfang_song16@163.com (Y.S.); ShainSQian@outlook.com (F.G.); hxqiuren@163.com (X.H.); zhaoyuzeng@shiep.edu.cn (Y.Z.)

**Keywords:** TiO_2_ nanofluid, Al_2_O_3_ nanofluid, sodium dodecyl benzene sulfonate, brass, corrosion

## Abstract

The corrosion behavior of brass in TiO_2_ and Al_2_O_3_ nanofluids using a simulated cooling water (SCW) as the base solution and sodium dodecyl benzene sulfonate (SDBS) as the dispersant was studied by electrochemical measurements and surface analysis in this paper. It was found that SDBS could be adsorbed on the brass surface to form a protective film and have a corrosion inhibition effect on brass in SCW. In the SCW-SDBS-TiO_2_ nanofluid, some negatively charged TiO_2_ nanoparticles were attached to the brass surface and no obvious SDBS adsorption film was found, and the SDBS in this nanofluid had almost no corrosion inhibition on brass. In the SCW-SDBS-Al_2_O_3_ nanofluid, the brass surface was covered by a uniformly distributed SDBS film containing some Al_2_O_3_ nanoparticles which were positively charged, and the corrosion inhibition of brass was significantly improved in this nanofluid. It is concluded that the adsorption of SDBS on the brass surface in nanofluids is related to the charge status of the nanoparticles, which makes brass have different corrosion resistance in various nanofluids.

## 1. Introduction

Nanofluids are a new type of energy-saving cooling medium, which refer to a uniformly dispersed medium formed by adding nanoparticles to a base solution. The addition of nano-sized particles can enhance the heat and mass transfer performance of the fluid [1,2]. Nanofluids have a good application prospect, such as in automobiles [3], solar energy [4] and air conditioners [5]. Replacing a traditional coolant with nanofluids in heat transfer systems can reduce energy consumption, decrease the size of the equipment and improve the working efficiency of systems [6]. However, the direct contact between the nanofluids and equipment may affect the corrosion behavior of metals. Some researchers have found that the addition of nanoparticles to the solution can promote metal corrosion [7,8]. The results of Fotowat et al. [9] indicated that alumina nanofluids had significant corrosive effects on both aluminum and copper, and the corrosion of copper was more severe. Bubbico et al. [10] found that the abrasion corrosion of the nanoparticles on the metal was not obvious, but the electrochemical corrosion on the metal surface under static conditions was relatively serious. The Brownian motion of nanoparticles in media can enhance the mass transfer process and accelerate the corrosion of metals in nanofluids [7]. However, other studies suggest that nanoparticles can inhibit metal corrosion. Nithiyanantham et al. [11] pointed out that nanoparticles can be incorporated into the oxide layer of the metal to reduce the corrosion rate of carbon steel.

In order to evenly disperse nanoparticles in the base solution, a certain amount of dispersants (surfactants) is often added to the nanofluids during the preparation [12,13,14]. Different dispersants not only have different influences on the stability and dispersibility of nanofluids, but also affect the corrosion behavior of metals to some extent [15]. The piperine surfactants which were synthesized by Tantawy et al. [16] can be chemically adsorbed on the steel surface, and have a corrosion inhibition effect on C1018 steel in a 3.5% NaCl solution. Sodium dodecyl benzene sulfonate (SDBS) is a common surfactant used in nanofluids, and it also exhibits a corrosion inhibition effect on metals at appropriate concentrations [17].

As a corrosion medium, a nanofluid contains both liquid and solid phases which can impact the corrosion behavior of metals. In this paper, TiO_2_ and Al_2_O_3_ nanoparticles with opposite charging properties were selected to prepare a TiO_2_ nanofluid and Al_2_O_3_ nanofluid, respectively, with simulated cooling water and commonly used SDBS as the base solution and dispersant. TiO_2_ and Al_2_O_3_ nanoparticles are both cheap and commonly used in nanofluids. In order to determine the effect of the nanoparticles’ charging properties on the aggressiveness of the nanofluids, the corrosion behavior of brass in these two kinds of nanofluids was studied and compared by electrochemical and surface analysis methods in this paper.

## 2. Experimental

### 2.1. Materials

The brass used in the experiments were type ASTM B111-C44300, and the composition is shown in Table 1. Brass plates were machined into 1 × 1 cm test pieces for the electrochemical experiments. Copper wire was welded onto the back of the working surface of the test piece, and the non-working surface was sealed with epoxy resin. Before each measurement, the working surface was ground step by step with emery papers ranging from 400 to 2000 mesh and rinsed with alcohol and distilled water.

### 2.2. The Experimental Medium

The base solution used for preparing the nanofluids was a simulated cooling water (SCW) which was composed of 7.5 mmol/L NaCl, 2 mmol/L NaHCO_3_, 3.5 mmol/L Na_2_SO_4_, 0.25 mmol/L MgSO_4_ and 0.5 mmol/L CaCl_2_. The pH value of SCW was adjusted to 8.0 with 0.01 mol/L HNO_3_ or 0.01 mol/L NaOH. The experimental TiO_2_ and γ-Al_2_O_3_ nanoparticles had an average particle size of 20 nm. The nanoparticles and dispersant SDBS were all from Aladdin Industries of China.

The TiO_2_ and Al_2_O_3_ nanofluids were prepared using sodium dodecyl benzene sulfonate (SDBS) as the dispersant [18]. First, the 500 mg/L nanoparticles and 500 mg/L SDBS were added into the prepared SCW. Then, the medium was stirred for 30 min by a magnetic agitator and further dispersed by ultrasound at a frequency of 45 kHz for 30 min, and finally a uniformly dispersed nanofluid was obtained. The zeta potential (ζ) was determined by a Melvin zeta potential analyzer (Nano—ZS90, Worcestershire, UK) to analyze the dispersion stability of the nanofluids. 

The experiments were performed in the following four media: simulated cooling water (SCW), simulated cooling water with 500 mg/L SDBS (SCW-SDBS), TiO_2_ nanofluid containing 500 mg/L SDBS (SCW-SDBS-TiO_2_ nanofluid) and Al_2_O_3_ nanofluid containing 500 mg/L SDBS (SCW-SDBS-Al_2_O_3_ nanofluid).

### 2.3. Electrochemical Measurements

The electrochemical measurements were conducted in a three-electrode system by using a CHI604E electrochemical workstation. A saturated calomel electrode (SCE) was used as the reference while a platinum plate was used as the auxiliary electrode. All tests were performed in an open system at 30 ± 1 °C. Electrochemical impedance spectroscopy (EIS) measurements were run at the open circuit potential. The test frequency range was from 100 kHz to 10 mHz, and the ac amplitude was 10 mV. The results of the EIS were fitted by the ZSimpWin software (ZSimpWin 3.60, EChem software, Michigan, MI, USA). The scanning rate for the measurement of the polarization curves was 1 mV/s. All tests were repeated more than three times for reliable results.

### 2.4. Characterization of the Metal Surface

The surface morphology of brass was observed by a JSM-7800 scanning electron microscope, and the composition of the brass surface was analyzed by EDS. 

## 3. Results and Discussion

### 3.1. Stability Analysis of TiO_2_ and Al_2_O_3_ Nanofluids

The stability of a nanofluid mainly depends on the charging state of the nanoparticles’ surface, and the repulsive or attractive force between the nanoparticles determines the dispersion or agglomeration of nanoparticles in the medium [19]. Surfactants can achieve a stable dispersion of nanofluids by altering the charging state of the nanoparticles [20,21,22]. Zeta potential (ζ) is usually used to judge the stability of nanofluids. When the absolute value of the zeta potential (|ζ|) of a nanofluid is higher than 30 mV, it can be considered that the nanofluid is stably dispersed [23,24].

Table 2 shows the zeta potentials of the TiO_2_ and Al_2_O_3_ nanofluids. The pH of the simulated cooling water for the experiment was about 8.0 and the surface of the TiO_2_ nanoparticles is negatively charged at this pH [25]. The ζ value of the TiO_2_ nanofluid without the surfactant was −19.8 mV. The Al_2_O_3_ nanoparticles are positively charged at this pH and the ζ value of the surfactant-free Al_2_O_3_ nanofluid was 3.28 mV. As a kind of anionic surfactant, the dissolved SDBS in the water solution can ionize and release in the form of anionic DBS^−^. In the SCW-SDBS-TiO_2_ nanofluid, SDBS can be adsorbed onto the TiO_2_ nanoparticles by van der Waals force, which makes the ζ value of the nanofluid more negative and increases the electrostatic repulsion between the TiO_2_ nanoparticles [26]. The |ζ| value of the SCW-SDBS-TiO_2_ nanofluid was 46.4 mV when the SDBS concentration was 500 mg/L. In the SCW-SDBS-Al_2_O_3_ nanofluid, the SDBS anions can be adsorbed on the surface of the positively charged Al_2_O_3_ nanoparticles by electrostatic adsorption, which improved the electrostatic repulsion between the nanoparticles as well. When the SDBS concentration was 500 mg/L, the |ζ| value of the SCW-SDBS-Al_2_O_3_ nanofluid was 40.9 mV. SDBS can disperse TiO_2_ and Al_2_O_3_ nanoparticles well in simulated cooling water.

### 3.2. EIS Analysis

The corrosion behavior of brass in different media was analyzed by EIS. Figure 1 shows the Nyquist plots of brass after five days of immersion in SCW, SCW-SDBS, the SCW-SDBS-TiO_2_ nanofluid and the SCW-SDBS-Al_2_O_3_ nanofluid. All the Nyquist plots of brass in the four media showed capacitive arcs, indicating that the brass corrosion in these media was mainly controlled by the charge transfer process. The Nyquist plots showed depressed capacitive arcs, which was mainly attributed to the dispersion effect caused by the uneven electrode surface roughness [27,28]. The EIS was fitted by using the equivalent circuit displayed in Figure 2 with two time constants, where R_s_ is the solution resistance, R_f_ and R_ct_ are the film resistance (due to the corrosion products or SDBS film on the metal surface) and the charge transfer resistance, respectively, and Q_dl_ and Q_f_ represent the double-layer capacitance and the film capacitance, respectively. The fitting results are displayed in Figure 1 (the solid line) and Table 3. For the purpose of obtaining better fitting results, the constant phase element Q was used instead of the pure capacitance when fitting [29].

Figure 1 shows that the fitting results were consistent with the experimental data. In SCW, two arcs in the Nyquist plot can be clearly distinguished, which correspond to the film resistance (R_f_) and film capacitance (Q_f_), the charge transfer resistance (R_ct_) and double-layer capacitance (Q_dl_), respectively. The R_f_ and Q_f_ are due to the corrosion products on the brass surface. In the Nyquist plots of the other media, the two arcs cannot be clearly separated, indicating that the two time constants are relatively close. It is shown in Table 3 that the R_ct_ values and R_f_ values of brass increased significantly after the addition of SDBS in SCW, which is due to the adsorption film of SDBS formed on the brass surface. The adsorption of SDBS can reduce the active sites on the brass surface and hinder the charge transfer [30]. Compared with the results in the SCW-SDBS medium, the R_ct_ and R_f_ values of the brass decreased significantly in the SCW-SDBS-TiO_2_ nanofluid, which indicates that the existence of TiO_2_ nanoparticles reduced the corrosion inhibition effect of SDBS on brass. In the SCW-SDBS-Al_2_O_3_ nanofluid, the R_f_ and R_ct_ values of the brass electrode were obviously higher than that in the SCW-SDBS medium, indicating that the Al_2_O_3_ nanoparticles enhanced the corrosion inhibition effect of SDBS on brass. 

### 3.3. Potentiodynamic Polarization Analysis

The polarization curves of brass after immersing in different media for five days are displayed in Figure 3. Table 4 exhibits the corrosion potential (*E*_corr_) and corrosion current density (*j*_corr_) obtained through the polarization curves. The results in Table 3 show that the *j*_corr_ value of the brass in SCW after five days of immersion is the largest (0.388 μA⋅cm^−2^), and it is the smallest (0.105 μA⋅cm^−2^) in the SCW-SDBS-Al_2_O_3_ nanofluid. The values of the Tafel slope (*b*_a_) were higher in both the SCW-SDBS-Al_2_O_3_ nanofluid and SCW-SDBS medium, indicating that the brass surfaces were well adsorbed by SDBS in these two media, which obviously suppressed the anodic dissolution of the brass electrode. The polarization current increases rapidly when the polarization potential is above 0.1 V, which corresponds to the desorption of SDBS on the brass surface. However, in the SCW-SDBS-TiO_2_ nanofluid, the *j*_corr_ value of the brass electrode is close to that in SCW. The shape of the anodic polarization curves in these two media is also similar, indicating that the surface states of brass in these two media may be similar. Besides, the *b*_a_ value of the brass electrode decreased in the SCW-SDBS-TiO_2_ nanofluid, indicating a decrease in the protection of the surface film. The change trend of *b*_c_ in the different media is consistent with *b*_a_, except in SCW, where concentration polarization might appear during the cathodic polarization process because of the large polarization current density. In the media containing SDBS, the cathodic polarization current density of the brass electrode at the same polarization value is relatively small, which should be due to the adsorption of SDBS on the metal surface, and reduces the effective area of the cathode. In addition, compared with the results in SCW, the corrosion potential of brass is negatively shifted, especially in the SCW-SDBS-Al_2_O_3_ nanofluid and SCW-SDBS medium, indicating that SDBS has a stronger inhibition on the cathode reaction.

### 3.4. Corrosion Products Characterization

Figure 4 shows the results of the SEM and EDS of the brass surfaces after five days of immersion in different media. As can be seen from Figure 4(a1), the brass surface was covered with loose corrosion products after five days immersion in SCW. According to the EDS results (Figure 4(a2)), the corrosion products were mainly composed of C, O and Zn, which should be the zinc compound Zn_5_(CO_3_)_2_(OH)_6_ [17]. The surface morphology of brass in SCW-SDBS is shown in Figure 4(b1). It can be seen that there were many aggregates attached to the brass surface. The EDS results (Figure 4(b2)) show that these aggregates mainly contained the elements C, O, S and Zn, among which the element C accounts for 59.24%, indicating that these aggregates were mainly SDBS and mixed with a small amount of the corrosion products of zinc. Alternatively, the surface that was not covered by the aggregates mainly contained the elements Cu, O and C (Figure 4(b3)), and the ratio of Cu to O was close to 2:1, implying the existence of the corrosion product Cu_2_O [31]. In the SCW-SDBS-TiO_2_ nanofluid, the surface morphology of brass was different from that in the previous two media. As shown in Figure 4(c1), no obvious accumulation of corrosion products and adsorption of SDBS were found on the brass surface, only some small particles were adsorbed and distributed on the surface. The EDS results (Figure 4(c2)) show that the brass surface with no particles was mainly composed of the elements C, O, Cu and Zn, with the atomic percentages of 14.55%, 22.89%, 59.23% and 3.33%, respectively, implying the existence of the corrosion product Cu_2_O. The EDS results of the particles attached to the brass surface show that the particles contained 51.06% of O and 22.80% of Ti (Figure 4(c3)), which should be the aggregate of the TiO_2_ nanoparticles. This indicates that the brass surface was mainly adhered to by the TiO_2_ nanoparticles in the SCW-SDBS-TiO_2_ nanofluid, and no obvious adsorption of SDBS was found. For specimens in the SCW-SDBS-Al_2_O_3_ nanofluid, it is shown in Figure 4(d1) that there was a relatively uniform adsorption film on the brass surface after five days of immersion. The EDS results (Figure 4(d2)) show that the brass surface contained 56.65% of C, 24.57% of O and 5.38% of S, which should correspond to the SDBS adsorption film on the brass surface. In addition, 4.39% of the Al element was also detected, indicating the existence of small amounts of Al_2_O_3_ nanoparticles mixed with the SDBS adsorption film. Comparing Figure 4(b1) and Figure 4(d1), it can be found that the adsorption film of SDBS on the brass surface was relatively dense and uniform in the SCW-SDBS-Al_2_O_3_ nanofluid, making the corrosion resistance of brass in this nanofluid better than that in the SCW-SDBS medium [17].

According to the above results, it can be seen that in SCW, the brass surface was mainly covered by loose corrosion products, which have poor protection for brass. In the SCW-SDBS medium, the positively charged brass surface [32] was protected by the adsorption film of SDBS, which improved the corrosion resistance of brass. In the SCW-SDBS-TiO_2_ nanofluid, a small amount of TiO_2_ nanoparticles were adsorbed on the brass surface and no obvious corrosion products and SDBS adsorption film were found. SDBS was not easy to adsorb on the brass surface in this nanofluid, which should be due to the competitive adsorption between the negatively charged TiO_2_ nanoparticles and the SDBS anions on the brass surface. The adhesion of the TiO_2_ nanoparticles inhibited the formation of an SDBS adsorption film on the brass surface, so that SDBS exhibited almost no corrosion inhibition effect on brass. In the SCW-SDBS-Al_2_O_3_ nanofluid, the Al_2_O_3_ nanoparticles are positively charged and would not be adsorbed on the brass surface which is also positively charged. An SDBS adsorption film was easy to form on the brass surface in this nanofluid. Furthermore, the positively charged Al_2_O_3_ nanoparticles could be adsorbed on the negatively charged SDBS film, improving the protection performance of the SDBS film on brass [32]. In addition, Al_2_O_3_ nanoparticles can also improve the critical micelle concentration of SDBS [17], therefore the SDBS adsorption film was not easy to aggregate on the brass surface. Therefore, brass has the best corrosion resistance in the SCW-SDBS-Al_2_O_3_ nanofluid among the four test media.

## 4. Conclusions

SDBS was adopted as a dispersant to prepare the TiO_2_ and Al_2_O_3_ nanofluids. The zeta potential of the two nanofluids containing 500 mg/L SDBS was −46.4 and −40.9 mV, respectively.

In SCW, the brass surface was covered by loose corrosion products and had poor corrosion resistance. In the SCW-SDBS medium, an SDBS adsorption film formed on the brass surface and improved the corrosion resistance of brass.

In the SCW-SDBS-TiO_2_ nanofluid, a small amount of TiO_2_ nanoparticles were adsorbed on the brass surface, and no obvious corrosion product and SDBS adsorption film were found. The adhesion of the negatively charged TiO_2_ nanoparticles on the brass surface inhibited the adsorption of the SDBS anions, which reduced the R_f_ and R_ct_ values of the brass electrode. In this nanofluid, the corrosion current density (*j*_corr_) of brass was larger, and the corrosion resistance of brass was close to that in SCW. 

In the SCW-SDBS-Al_2_O_3_ nanofluid, the *R*_f_ and *R*_ct_ values of the brass electrode were the maximum and the corrosion current density (*j*_corr_) was the minimum. The brass surface was covered with a relatively dense SDBS adsorption film containing a small number of Al_2_O_3_ nanoparticles. The positively charged Al_2_O_3_ nanoparticles promoted the formation of a denser SDBS adsorption film, which obviously improved the corrosion resistance of brass.

## Figures and Tables

**Figure 1 nanomaterials-10-01046-f001:**
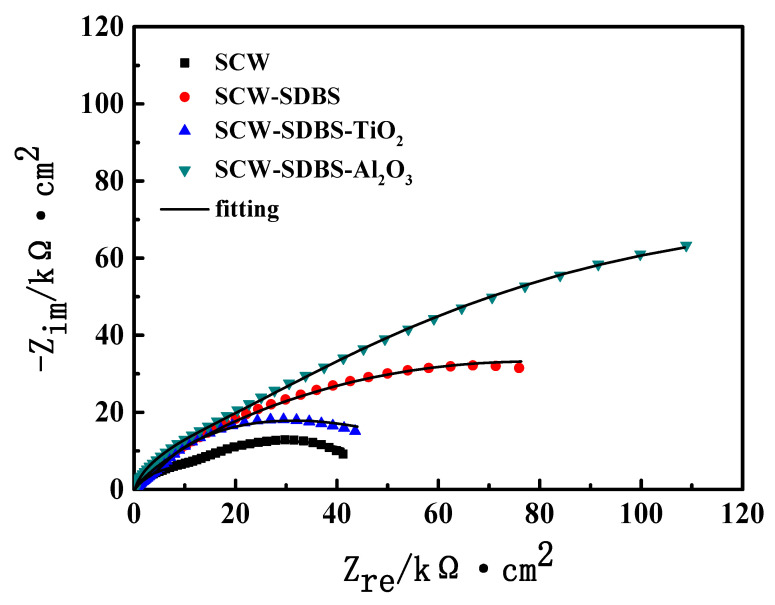
Nyquist plots of brass immersed in different media for 5 days.

**Figure 2 nanomaterials-10-01046-f002:**
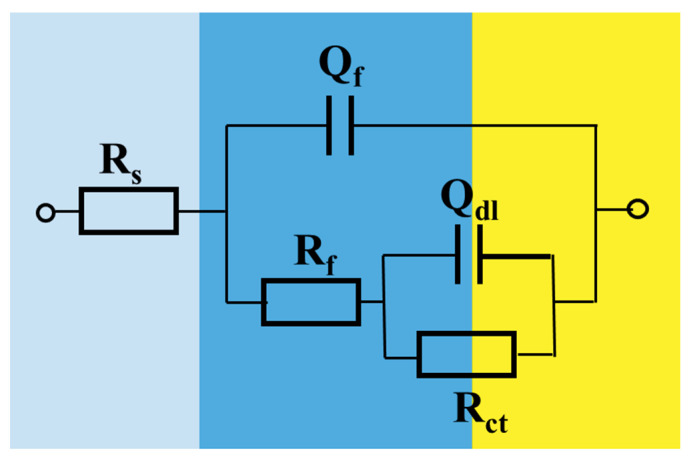
Equivalent circuit for fitting electrochemical impedance spectroscopy (EIS).

**Figure 3 nanomaterials-10-01046-f003:**
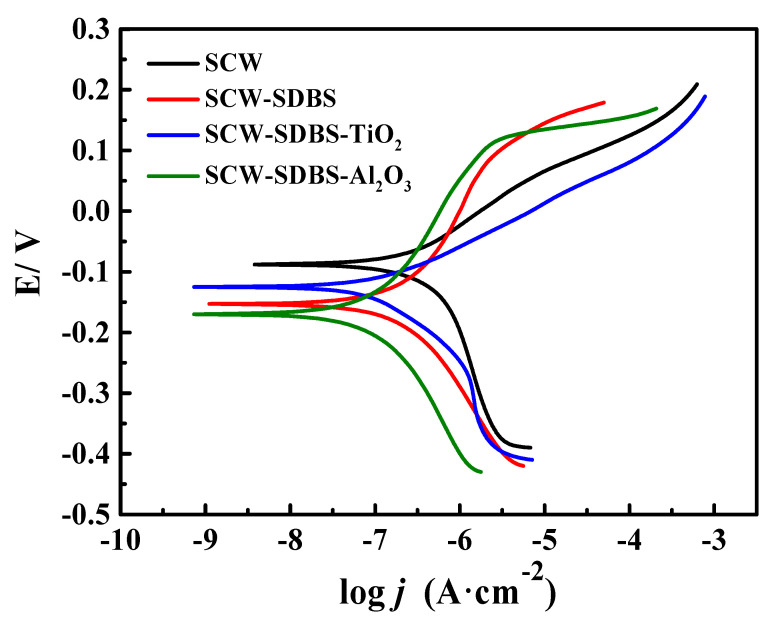
Potentiodynamic polarization curves of the brass electrode after immersion for 5 days in different media.

**Figure 4 nanomaterials-10-01046-f004:**
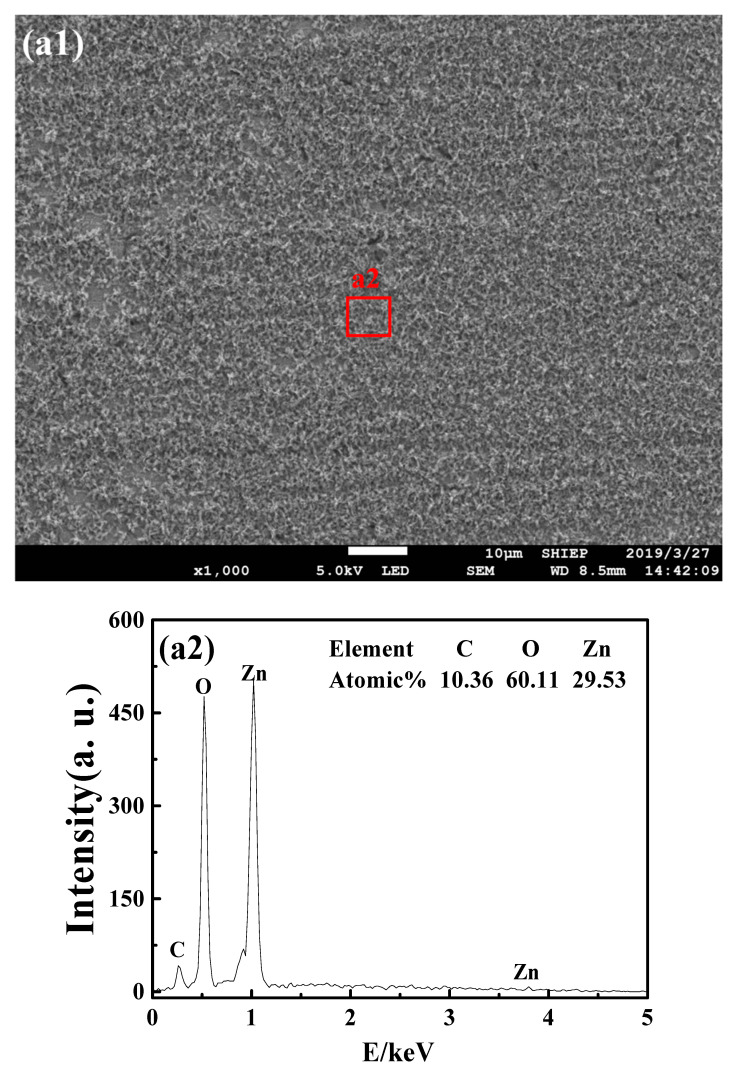
SEM photos and EDS results of the brass surface after immersion of 5 days in different media. (**a**) Simulated cooling water (SCW); (**b**) SCW-sodium dodecyl benzene sulfonate (SDBS); (**c**) SCW-SDBS-TiO_2_ nanofluid; and (**d**) SCW-SDBS-Al_2_O_3_ nanofluid.

**Table 1 nanomaterials-10-01046-t001:** The composition of brass (ASTM B111-C44300) (wt %).

Elements	Cu	Sn	Fe	Pb	As	Bi	P	Zn
Contents	69.9	0.90	0.10	0.05	0.04	0.002	0.01	the rest

**Table 2 nanomaterials-10-01046-t002:** The zeta potential (ζ) (mV) of the different nanofluids.

*C*_SDBS_ (mg/L)	0	500
SCW-SDBS-TiO_2_ nanofluid	−19.8	−46.4
SCW-SDBS-Al_2_O_3_ nanofluid	3.28	−40.9

**Table 3 nanomaterials-10-01046-t003:** Fitting results of the EIS in Figure 1.

Test Media	*R* _s_	*R* _f_	*Q* _f_	*n* _1_	*R* _ct_	*Q* _dl_	*n* _2_
	Ω·cm^2^	kΩ·cm^2^	*Y*_f_(μS·s^n^cm^−2^)		kΩ·cm^2^	*Y*_dl_(μS·s^n^cm^−2^)	
SCW	162.1	10.59	28.71	0.80	44.70	53.57	0.58
SCW-SDBS	173.8	21.01	23.44	0.74	163.5	32.25	0.85
SCW-SDBS-TiO_2_	131.7	13.58	26.57	0.75	62.61	48.82	0.65
SCW-SDBS-Al_2_O_3_	150.0	27.27	18.94	0.87	266.1	27.17	0.75

**Table 4 nanomaterials-10-01046-t004:** Electrochemical parameters of brass in the four media.

Test Media	*E* _corr_	*b* _a_	*b* _c_	*j* _corr_
	(mV)	(mV dec^−1^)	(mV dec^−1^)	(μA·cm^−2^)
SCW	−87	103	250	0.388
SCW-SDBS	−154	188	174	0.161
SCW-SDBS-TiO_2_	−125	95	157	0.302
SCW-SDBS-Al_2_O_3_	−171	192	197	0.105

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
