# Peer review of "Comparison of the Corrosion Behavior of Brass in TiO2 and Al2O3 Nanofluids"

_nanomaterials, 2020, doi:10.3390/nano10061046_

Round 1

Reviewer 1 Report

  The article is fairly well written. Minor grammar and typing problems can be easily fixed. The presentation of the research and the findings are correct. It would be good if the authors could justify what indicated the choice of nanoparticles, the dispersant. How was the nanoparticle concentration applied in this article chosen? Why was the effect of the nanofluid studied after just 5 days? It would be interesting to analyze the effect of both concentration and time.

Reviewer 2 Report

The manuscript could be published if revised according to the comments presented below:

  1. Line 30 - This sentence needs additional citations - "...efficiency of system".
  2. Lines 52-53 - This sentence is not very clear and needs to be rewritten.
  3. Line 113 - "...neutral medium" is not correct since the pH value is 8.
  4. Table 2 - The role of concentration of 150 mg/l SDBS is not clear.
  5. Line 137 - "... Fig. 1 and Table 2" seems to be "...Fig. 1 and 2".
  6. Fig. 1 and Fig. 2 - It seems that EIS data of SCW has different behavior (maybe 2 arcs) which is not commented. Is the equivalent circuit on Fig 2 applied for all samples?
  7. Line 159 - "Table 3..." seems to be Table 4.
  8. Fig. 3 - The blue and red curves seem to have close Jcorr values but differ in Table 4? More clarity is needed.
  9. Line 186 - "...Fig. 4(a1)" seems to be Fig. 4(a2)?
  10. Line 202 - "CuO2"???

Round 2

Reviewer 2 Report

The manuscript seems to be more suitable for publishing compared to its first version. However, one more question appears for me which needs more clarity.

Table 4 - The authors give other "bc" values for three model media (SCW-SDBS; SCW-SDBS-TiO2; SCW-SDBS-Al2O3, respectively) compared to the first version. In my opinion this will affect also the jcorr values which are not changed. Please, comment the case.
